# The Cow’s Milk-Related Symptom Score (CoMiSS^TM^) in Presumed Healthy Egyptian Infants

**DOI:** 10.3390/nu16162666

**Published:** 2024-08-12

**Authors:** Wael A. Bahbah, Nienke Knockaert, Heba M. S. El Zefzaf, Koen Huysentruyt, Yvan Vandenplas

**Affiliations:** 1Department of Pediatrics, Faculty of Medicine, Menoufia University, Shebin El-Kom 32511, Egypt; wael_bahbah@yahoo.com (W.A.B.); heba_magdy4@yahoo.com (H.M.S.E.Z.); 2KidZ Health Castle, UZ Brussel, Vrije Universiteit Brussel (VUB), 1090 Brussels, Belgium; knockaert.nienke@gmail.com (N.K.); koen.huysentruyt@uzbrussel.be (K.H.)

**Keywords:** Cow’s Milk-Related Symptom Score (CoMiSS), cow’s milk allergy (CMA), infants, awareness

## Abstract

Background: The Cow’s Milk-Related Symptom Score (CoMiSS) was created as an awareness tool for cow’s milk-related symptoms. After different trials, a score of ≥10 was selected to raise awareness. The CoMiSS in healthy infants needs to be determined because the score does not return to 0 during a diagnostic elimination diet. This study aims to establish normal values in healthy Egyptian infants. Methods: In this prospective cross-sectional study, pediatricians determined the CoMiSS in healthy infants ≤ 12 months. Infants seeking medical help due to cow’s milk allergy (CMA) symptoms and infants with any known or suspected diseases, preterm delivery, medication, or food supplements were excluded. Results: A total of 808 infants were included with a median (Q1; Q3) age of 7 (3;10) months (50.7% boys). The median (Q1; Q3) CoMiSS was 5 (5;6). The 95th percentile was 7. There was no significant difference in the median CoMiSS according to gender (*p* = 0.621) or due to breastfeeding exclusively (*p* = 0.603). A significant difference was seen in the CoMiSS according to age, although all the age categories had a median CoMiSS of 5. Conclusions: This study revealed the median CoMiSS is 5 in presumed healthy Egyptian infants aged 0–12 months. The CoMiSS was not dependent on feeding. The determination of the CoMiSS in healthy infants allows for the determination of a cut-off under which CMA is unlikely, and a cut-off to raise awareness of CMA, thereby preventing under- and overdiagnosis. Since the median CoMiSS was not different in European infants, the outcome suggests that the CoMiSS may be a reliable awareness tool for CMA independent of ethnicity. However, additional studies are needed to confirm the previous hypothesis.

## 1. Introduction

Infants with gastrointestinal (GI) symptoms (vomiting, regurgitation, loose and watery stools, constipation), respiratory tract symptoms (wheezing, chronic cough), and general symptoms (poor growth, infantile colic, and persistent distress) are frequently seen by health care professionals (HCPs). Although these symptoms generally affect 15–20% of infants, they can also be linked to cow’s milk allergy (CMA) diagnosis [1].

Identifying CMA remains difficult in clinical practice. The prevalence of CMA is variable in the literature due to different assessment methods [2]. Reviews have reported a prevalence between 1.9 and 4.9% [3], although according to the EuroPrevall study and to the European Society of Pediatric Gastroenterology, Hepatology and Nutrition (ESPGHAN), the prevalence is <1% [3]. An Egyptian study estimated the prevalence of CMA at 3.4%, confirmed by an oral food challenge, which is the golden diagnostic standard [4].

In 2015, a group of European experts developed the Cow’s Milk-Related Symptom Score (CoMiSS^TM^), which is a score that is intended to raise awareness of the healthcare professional regarding the fact that the symptomatology presented may be indicative of a diagnosis of CMA. The CoMiSS generates a score based on the crying duration, amount of regurgitation, stool score, skin manifestations, and respiratory symptoms [1]. The CoMiSS uses a scale from 0 to 33 and was updated in 2022 (Table 1). A score of ≥10 should raise awareness of CMA. Overdiagnosis, which can cause a tremendous economic burden, can be prevented since at least two symptoms need to be present to reach a score of 10 or above. Underdiagnosis, which can lead to growth faltering and nutritional risk, can be prevented by using the score as an assessment to create more awareness for the diagnosis of CMA [3,5].

Experience has shown that the CoMiSS does not return to 0 during a diagnostic elimination diet, indicating that healthy infants also present some degree of the manifestations included in the CoMiSS. It was also observed that if, during an elimination diet, the initial CoMiSS was reduced to ≤6, the oral food challenge was positive in 80% of the cases [1].

In order to be better able to estimate the efficacy of a diagnostic elimination diet, the CoMiSS needs to be determined in healthy infants. In healthy European infants aged 0 to 12 months, the CoMiSS was reported to have a median score of 3 [6,7]. To have a better overview of the CoMiSS in healthy infants globally, more countries need to be examined.

This study will inform local Egyptian healthcare professionals of the baseline CoMiSS in presumed healthy Egyptian infants, improving clinical decision-making. Besides the local impact, this study will also show a similarity or a difference to existing data from European cohorts. We hypothesize that the median CoMiSS in healthy Egyptian infants will be ≤6, supporting previous research.

## 2. Materials and Methods

### 2.1. Setting and Participants

A prospective cross-sectional study was conducted on 991 healthy Egyptian infants from June to October 2023. Participants aged 0–12 months were selected randomly from a discrete rural and urban region to reflect Egyptian infants. Exclusion criteria included infants with any known or suspected diseases, infants seeking medical consultation due to symptomatology, preterm delivery (<37 weeks), intake of food supplements, or use of any medication except for the recommended vitamins. Trained pediatricians assessed CoMiSS during regular planned follow-up visits or during vaccination sessions through interviews in the native language. Brussels Infant and Toddlers Stool Scale (BITSS) was printed, and mothers were asked to point to the most appropriate picture and its score was documented to avoid mistakes about the parental perception of symptoms. We obtained ethical approval from the Institutional Review Board (IRB) of the Menoufia Faculty of Medicine (ID number: 6/2023 PEDI 6). This research was performed in accordance with the Declaration of Helsinki. Informed consent was obtained from all the legally authorized representatives of participants included in the study. There was no funding for this study.

### 2.2. Statistical Analysis

The statistical analyses in this study were conducted using SPSS29, with a predetermined significance level of *p* < 0.05 indicative of statistical significance. Categorical variables were analyzed through the chi-square test, while continuous variables were analyzed using the Mann–Whitney U or Kruskal–Wallis test. Normality was checked using the Kolmogorov–Smirnov test, presenting normally distributed values as mean (Standard Deviation (SD)) and non-normally distributed values as median and interquartile ranges (IQRs). A regression analysis was used to examine the impact of age, gender, feeding type, and country on CoMiSS.

Age-dependent analyses included comparisons across different categories. We compared 0–2 months, 2–4 months, 4–6 months, 6–8 months, 8–10 months, and 10–12 months, referred to as the first age category analysis in this scientific paper. Furthermore, comparisons were made between the broader categories of 0–6 months and 6–12 months, referred to as the second age category analysis.

## 3. Results

### 3.1. Population Description

During the research, we collected information from 991 Egyptian infants. Of these infants, 131 were excluded due to an age above 12 months and 52 due to the incorrect notation of the birth date. A total of 808 infants were included in the final analysis with a median (Q1; Q3) age of 7 (3; 10) months. There were 411 (50.9%) boys included and 397 (49.1%) girls. There was no significant difference in age between the boys and girls (*p* = 0.828). Of these infants, 6.2% were exclusively breastfed, 56.3% of the infants were breastfed (total: mixed + exclusively), 78.1% were formula-fed, and 66.5% had started solid feeding. The patient characteristics are shown in Table 2.

### 3.2. Overall Values of CoMiSS

The median (Q1; Q3) CoMiSS was 5 (5; 6) (Figure 1). The 95th percentile was 7. None of the infants had a CoMiSS of ≥10. There was no significant difference in the CoMiSS according to gender (*p* = 0.626) (Appendix A Figure A1), due to breastfeeding exclusively (*p* = 0.198) (Figure A2), breastfeeding (total: mixed + exclusively) (*p* = 0.430), formula feeding (*p* = 0.886), or when complementary feeding was started (*p* = 0.202) (Table A1).

A significant difference was seen in the CoMiSS in the first age category analysis (*p* = 0.02) (Figure 2), but not in the second age category analysis (*p* = 0.840) (Figure 3). A median CoMiSS of 5 was seen in all the age categories (Table 3). A higher 25th percentile of 5 was seen in the 4–6 months, 8–10 months, and 10–12 months groups compared to a 25th percentile of 4 in the other age categories.

### 3.3. Subscores of CoMiSS

#### 3.3.1. Crying Score

The median crying scores were significantly different in the first age category analysis (*p*
< 0.001) and in the second age category analysis (*p* = 0.002) (Table A1). An increase in the median (Q1; Q3) crying score is seen with increasing age with a median of 0 (0; 1) in the 0–6 months group and a median of 1 (0; 2) in the 6–12 month group. The most frequent crying score (54.5%) was 0, which represents crying ≤1 h per day. Only 23% of the infants cried more than 1.5 h a day. This occurred mainly in the older age categories (29.4% in the 6–8 months group and 27.7% in the 8–10 months group). There was no significant difference in the crying scores according to gender (*p* = 0.922).

#### 3.3.2. Regurgitation Score

We observed a significant difference in regurgitation in the first age category analysis (*p* = 0.005) but not in the second age category analysis (*p* = 0.099) (Table A1). The most frequent regurgitation score was 1 (≥3 to ≤5 regurgitations a day), present in 56.2% of the infants. A score of 0, representing 0–2 regurgitations a day, occurred most in the 6–8 months group (50.4%). The highest regurgitation score reported in the included infants was 2 (regurgitation more than five times in less than half of the feedings), mostly in the 4–6 months group (14%), not reported in the 8–10 months group (0%), and barely in the 10–12 months group (0.9%). No significant difference was observed in the regurgitation score according to breastfeeding (*p* = 0.401), formula feeding (*p* = 0.655), solid feeding introduction (*p* = 0.648), or gender (*p* = 0.607).

#### 3.3.3. Stool Score

The median stool score was significantly different across both age category analyses (*p* < 0.001) (Table A1). Most of the infants had loose stools (91.6%). Only 1% had watery stools, which were more frequent in the 4–6 months age group (2.5%) and not in the 0–2 months and 10–12 months group. There was no significant difference in the stool score due to breastfeeding exclusively (*p* = 0.081) or according to gender (*p* = 0.171).

#### 3.3.4. Skin Symptoms

No infants were reported to have eczema, urticaria, or angioedema.

#### 3.3.5. Respiratory Score

There was no significant difference in the respiratory score according to both age category analyses (*p* = 0.298, *p* = 0.335) (Table A1). No respiratory symptoms were reported in 96.2% of the infants. There was no significant difference in the respiratory score according to gender (*p* = 0.932).

## 4. Discussion

In presumed healthy Egyptian infants, the median (Q1; Q3) CoMiSS was 5 (5; 6). The 95th percentile was 7. The overall median CoMiSS was the same in all of the age categories. Comparable to this study, the European studies showed no difference in the median CoMiSS in the 0–6 months age group compared to the 6–12 months age group. The studies in European infants showed a median CoMiSS of 3 [6].

Crying (*p* < 0.001), regurgitation (*p* = 0.004), and stool scores (*p* < 0.001) were significantly different across the different age categories. The other factors of the CoMiSS were not age-dependent. The highest crying score was observed in the 6–8 months group; this is contradictory to the literature showing a decrease in crying after the age of 6 weeks [8].

The highest regurgitation score occurred in the 4–6 months group, was 0 in the 8–10 months group, and was barely seen in the 10–12 months group. This is in line with the literature showing a peak of regurgitation at 4 months and a reduction with increasing age [9,10]. Most of the infants had loose stools, which means that they had a stool score of 4 in the CoMiSS. This contributes substantially to the higher median CoMiSS in Egyptian infants compared to European infants. A higher median CoMiSS may also suggest that help may be sought more often for Egyptian infants because of different manifestations when these symptoms are more important than compared to European infants.

Although the Egyptian infants had a higher median CoMiSS than the European infants, none of the Egyptian infants had urticaria or eczema. That means that all of these infants had a skin score of 0 in the CoMiSS. This could suggest that pediatricians in Egypt do not consider symptoms such as eczema or urticaria as occurring in presumed healthy infants. The CoMiSS has become part of the diagnostic work-up if infants are suspected to suffer from CMA [11,12,13].

A limitation of the CoMiSS is that it depends on the parents’ recollective memory during the consultation, which may cause an exaggeration of the symptoms compared to daily prospective or real-time collection. Furthermore, infants visiting welfare clinics or visiting regular follow-up consultations were selected, creating a potential selection bias [1,14]. In order to improve these limitations, research is currently ongoing involving collecting data from other ethnic groups, and a longitudinal study in a group of infants has been performed as well. A “Pre-CoMiSS” tool for parents to collect information prospectively is under development. It will be equally important to determine and compare the CoMiSS values in infants with diagnosed CMA according to their ethnic origin, age, and other variables.

## 5. Conclusions

The median CoMiSS in healthy Egyptian infants is 5. This result can be helpful in assessing CMA in infants. When there is a high CoMiSS, and particularly a score of ≥10, an elimination diet and oral food challenge are needed to confirm the presumed diagnosis. It is important to note that the CoMiSS is an assessment tool, not a diagnostic tool for CMA. Future research has been set up to examine the median CoMiSS in healthy infants worldwide. Longitudinal research could give us a more thorough perspective on how the CoMiSS progresses with age, the emergence of symptoms, and could help ensure its applicability across different populations. Additionally, it is crucial to provide proper training on applying the CoMiSS to avoid potential misapplications in the clinical environment.

## Figures and Tables

**Figure 1 nutrients-16-02666-f001:**
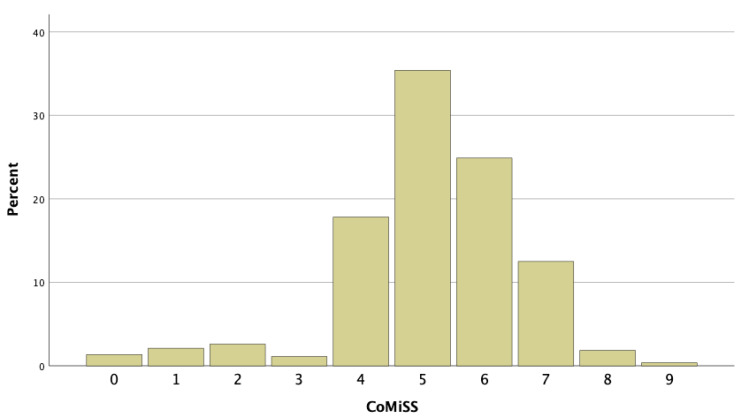
Distribution of overall CoMiSS in Egyptian infants.

**Figure 2 nutrients-16-02666-f002:**
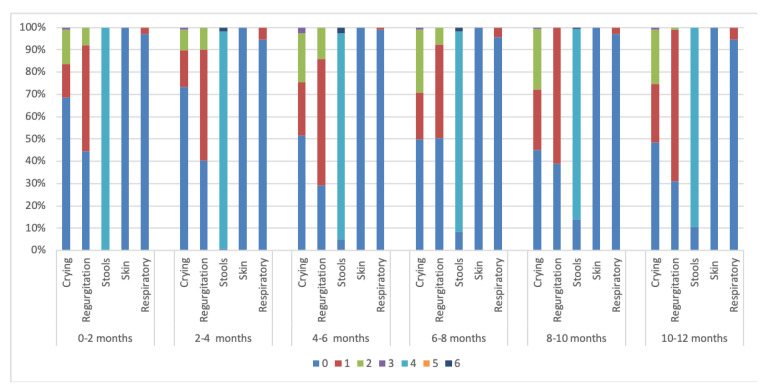
Distribution of CoMiSS according to the first age category analysis in Egypt.

**Figure 3 nutrients-16-02666-f003:**
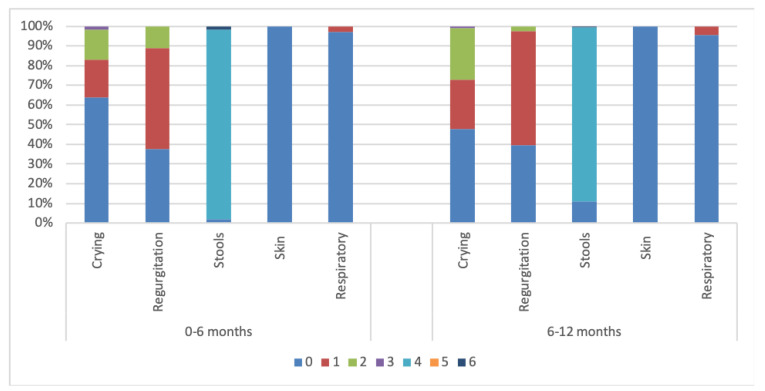
Distribution of CoMiSS according to the second age category analysis in Egypt.

**Table 1 nutrients-16-02666-t001:** The CoMiSS [1].

Symptoms	Score				
Crying	0	≤1 h/day			
* assessed by parents	1	1–1.5 h/day			
and without any obvious	2	1.5–2 h/day			
cause ≥1 week duration	3	2 to 3 h/day			
	4	3 to 4 h/day			
	5	4 to 5 h/day			
	6	≥5 h/day			
Regurgitation	0	0–2 episodes/day			
* ≥1 week duration	1	≥3–≤5×of volume < 5 mL			
	2	>5 episodes of >5 mL			
	3	>5 episodes of ±half of the feed in			
	4	Continuous regurgitations of small volumes > 30 min after each feed			
	5	Regurgitation of half to complete volume of a feed in at least half of the feeds			
	6	Regurgitation of the complete feed after each feeding			
Stools	4	Hard stools			
Brussels Infant and	0	Formed stools			
Toddlers Stool	4	Loose stools			
Scale (BITSS)	6	Watery stools			
* ≥1 week duration					
Skin symptoms	0–6	Atopic eczema	Head–neck–trunk	Arms–legs–hands–feet	
		Absent	0	0	
		Mild	1	1	
		Moderate	2	2	
		Severe	3	3	
	0–6	Urticaria (0: no, 6: yes)			
Respiratory	0	No respiratory symptoms			
symptoms	1	Slight symptoms			
	2	Mild symptoms			
	3	Severe symptoms			

Legend: * ≥1 week duration.

**Table 2 nutrients-16-02666-t002:** Patient characteristics.

	*n* (%)
Total	808 (100%)
Boys	411 (50.9%)
Girls	397 (49.1%)
Exclusively breastfed	50 (6.2%)
Breastfeeding (total)	454 (56.2%)
Formula feeding	628 (78.1%)
Feeding solids	536 (66.5%)
Median age in months	7 (3.12)
0–2 months	99 (12.3%)
2–4 months	109 (13.55)
4–6 months	120 (14.9%)
6–8 months	119 (14.7%)
8–10 months	144 (17.85)
10–12 months	217 (26.8%)
0–6 months	328 (40.6%)
6–12 months	480 (59.4%)

Legend: *n*: number of infants.

**Table 3 nutrients-16-02666-t003:** Descriptive outcomes in healthy Egyptian infants.

	P5	P25	Median	P75	P95	Min–Max
Total (*n* = 808)	2	5	5	6	7	0–9
Boys (*n* = 410)	3	5	5	6	7	0–9
Girls (*n* = 398)	2	4	5	6	7	0–9
Excl. breastfeeding (*n* = 50)	4	4	5	6	7	4–9
Breastfeeding (total, *n* = 455)	2	5	5	6	7	0–9
Formula feeding (*n* = 628)	2	5	5	6	7	0–9
Feeding solids (*n* = 536)	1	5	5	6	7	0–9
0–2 months (*n* = 99)	4	4	5	6	7	4–9
2–4 months (*n* = 109)	4	4	5	6	7	3–9
4–6 months (*n* = 120)	2	5	5	6	7	0–9
6–8 months (*n* = 120)	2	4	5	6	7	0–8
8–10 months (*n* = 144)	1	4	5	6	7	0–8
10–12 months (*n* = 216)	1	5	5	6	7	0–8
0–6 months (*n* = 328)	4	5	5	6	7	0–9
0–12 months (*n* = 480)	1	4	5	6	7	0–8

Legend: As the table represents the percentile curves, further statistical analysis is not possible.

## Data Availability

For access to the information/database presented in this research, please contact the corresponding author.

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
