# Peer review of "The Cow’s Milk-Related Symptom Score (CoMiSSTM) in Presumed Healthy Egyptian Infants"

_nutrients, 2024, doi:10.3390/nu16162666_

Round 1
Reviewer 1 Report
Comments and Suggestions for Authors
REVIEW for the journal Nutrients (ISSN 2072-6643)
Article “The Cow’s Milk-Related Symptom Score (CoMiSSTM) in presumed healthy Egyptian infants”
Manuscript ID: nutrients-3123866
Authors: Wael A Bahbah, Nienke Knockaert, Heba M.S. El Zefzaf, Koen Huysentruyt, Yvan Vandenplas
Brief summary. The study addresses an important tool, the Cow’s Milk-Related Symptom Score (CoMiSS), used to identify cow’s milk-related symptoms, which is significant for early detection and management of cow’s milk allergy (CMA).
Comments
1. Abstract.
- I suggest that the authors provide a summarizing sentence at the end to highlight the significance of their study.
2. Introduction.
- Here is a suggestion to improve the introduction
- Elaborate more on future research directions: Provide more detail on what specific aspects should be studied further.
- Formulate a clear hypothesis.
- Refine the research objectives, amplifying them with potential impact, from the local to the more global.
- I don't think it was a good idea to include the entire table from reference 12 (Table 1) in the introduction; it would be enough to comment on it.
3. Methodology.
The prospective cross-sectional design is appropriate for establishing normative data. The study meticulously excludes infants with known diseases, preterm delivery, medication, or food supplements, ensuring a more accurate assessment of CoMiSS in healthy infants. The inclusion of 808 infants provides a robust data set, enhancing the reliability of the findings. With a nearly equal representation of boys and girls, the study ensures gender-related findings are more generalizable.
- I have concerns about the limited scope of this study: It is unclear what implications this study may have for other populations due to nutritional and genetic differences that may influence CoMiSS?
- Line 85. This needs an explanation: “mean (SD)” ??
4. Results
- 3.1. Population description.
Data on sample size and structure should be provided in the methodology section.
- Table 2. The sample size must be denoted by the symbol "n" (populations - "N").
5. The conclusion
The article's findings regarding the median CoMiSS in healthy Egyptian infants offer a valuable reference point for assessing CMA. However, the implications of a median score of 5, the appropriateness of the ≥ 10 threshold, and the role of CoMiSS as an assessment tool necessitate careful consideration. Future research should focus on broader, longitudinal studies to validate these findings and ensure their applicability across different populations. Proper training on the use of CoMiSS is also essential to prevent its misuse in clinical settings.
6. The reference list
should be carefully reviewed to ensure it complies with the journal’s specific requirements.
Cross-sectional data provide a snapshot in time but do not capture changes in CoMiSS as infants grow. Longitudinal studies could provide more comprehensive insights into how CoMiSS evolves with age and the onset of symptoms.
In my opinion, the authors do not emphasize enough that future research should include a diverse range of populations to ensure the findings are applicable globally. This would involve examining CoMiSS in different ethnic groups, socioeconomic backgrounds, and dietary practices to develop a more universal understanding of normal CoMiSS values.
Sincerely, Reviewer.
Author Response
Reviewer 1.
I suggest that the authors provide a summarizing sentence at the end to highlight the significance of their study.
We added a sentence: CoMiSS was not dependent on feeding. Since the median CoMiSS was not different in European infants, the outcome suggests that CoMiSS may be a reliable awareness tool for CMA independent of ethnicity. However, additional studies are needed to confirm the previous hy-pothesis.
Introduction
Elaborate more on future research directions: Provide more detail on what specific aspects should be studied further.
We included more information regarding future research, but did this at the end of the paper. For the personal information of the reviewer: similar data have bene obtained in Indonesian, Brazilian and Mexican children. We have now CoMiSS data in over 2000 presumed healthy infants.
Formulate a clear hypothesis.
A clear hypothesis sentence was added at the end of the Introduction-section.
Refine the research objectives, amplifying them with potential impact, from the local to the more global.
This aspect is included in the added sentences in the Conclusion section.
I don't think it was a good idea to include the entire table from reference 12 (Table 1) in the introduction; it would be enough to comment on it.
Many health care professionals still do not know CoMiSS. And since Nutrients is only an "on-line" Journal, the inclusion of the CoMiSS as a full table does not increase the use of paper. Therefore, we prefer to keep the table.
Methodology.
The prospective cross-sectional design is appropriate for establishing normative data. The study meticulously excludes infants with known diseases, preterm delivery, medication, or food supplements, ensuring a more accurate assessment of CoMiSS in healthy infants. The inclusion of 808 infants provides a robust data set, enhancing the reliability of the findings. With a nearly equal representation of boys and girls, the study ensures gender-related findings are more generalizable.
We thank the reviewer for the positive comment.
I have concerns about the limited scope of this study: It is unclear what implications this study may have for other populations due to nutritional and genetic differences that may influence CoMiSS?
This aspect is now also included in the Conclusion section.
Line 85. This needs an explanation: “mean (SD)” ??
We clarified.
Results
3.1. Population description.
Data on sample size and structure should be provided in the methodology section.
Number of included infants is in the methods section. The reasons for exlcusion belong typically in the Result section.
Table 2. The sample size must be denoted by the symbol "n" (populations - "N").
Done
The conclusion
The article's findings regarding the median CoMiSS in healthy Egyptian infants offer a valuable reference point for assessing CMA. However, the implications of a median score of 5, the appropriateness of the ≥ 10 threshold, and the role of CoMiSS as an assessment tool necessitate careful consideration. Future research should focus on broader, longitudinal studies to validate these findings and ensure their applicability across different populations. Proper training on the use of CoMiSS is also essential to prevent its misuse in clinical settings.
As said, we are accumulating infirmation on CoMiSS in presumed helathy infants, worldwide. And a paper on a longitudinal follow-up of a group of infants is now in review.
The reference list
should be carefully reviewed to ensure it complies with the journal’s specific requirements.
Done
Cross-sectional data provide a snapshot in time but do not capture changes in CoMiSS as infants grow. Longitudinal studies could provide more comprehensive insights into how CoMiSS evolves with age and the onset of symptoms.
In my opinion, the authors do not emphasize enough that future research should include a diverse range of populations to ensure the findings are applicable globally. This would involve examining CoMiSS in different ethnic groups, socioeconomic backgrounds, and dietary practices to develop a more universal understanding of normal CoMiSS values.
We agree with the comment of the reviewer. As already mentioned, all these aspect have bene considered and the research is being done at this moment, and some of this information is already "in review" to be published soon.

Reviewer 2 Report
Comments and Suggestions for Authors
This is a study presenting The Cow's Milk-Related Symptom Score in Egyptian infants. Introduction justifies the study, methods which were applied are appropriate. In my opinion, the presented article contains a serious deficiency in the analysis of susceptibility to cow's milk allergy in the studied infants. Without a comparison of the CMA and CoMiSS results, your study has only partial value and at best can be classified as a short communication when corrected.
The most important part in your study is statistical analysis. You need to present it in a separate table or on the figures which you discuss because when you only present it in the text it is really hard to look at it.
Besides:
Line 176-178
You need to unify how you present P value
Author Response
Reviewer 2
This is a study presenting The Cow's Milk-Related Symptom Score in Egyptian infants. Introduction justifies the study, methods which were applied are appropriate. In my opinion, the presented article contains a serious deficiency in the analysis of susceptibility to cow's milk allergy in the studied infants. Without a comparison of the CMA and CoMiSS results, your study has only partial value and at best can be classified as a short communication when corrected.
In order to raise "justified awareness for cow's milk allergy", it is of major relevance to have information of the used awareness tool in healthy infants. This is exactly what is done in this paper in Egyptian infants. In the mean time, for the personal information of the reviewer, a similar collection of data has been done in Euroepan (already published), Indonesian, Mexican and Brazilian infants.
The most important part in your study is statistical analysis. You need to present it in a separate table or on the figures which you discuss because when you only present it in the text it is really hard to look at it.
The median CoMiSS in the different categories are mentioned in Table 3.
Besides: Line 176-178. You need to unify how you present P value
Done

Round 2
Reviewer 1 Report
Comments and Suggestions for Authors
I thank the authors for the improvements made to the manuscript and hope that my observations, which they have fully considered, have contributed to enhancing its quality.
Author Response
We thank the reviewer for his reply and nice comment.
Reviewer 2 Report
Comments and Suggestions for Authors
I'm disappointed your response. I guess you don't want to correct this article more profoundly, but really, adding an additional table that doesn't require additional research is probably not a big challenge. Table 3 that you indicate doesn't have a p-value. If you want your article to be cited, you need to better highlight the statistical differences that you obtained.
If it is not possible to conduct the additional research that I mentioned earlier (CMA), it would be fair to write in the section about the weaknesses of your work that the best confirmation of your studies would be an additional CMA analysis, but due to financial, logistical, etc. constraints, this is not possible
Author Response
Dear Reviewer
We did carefully discuss and consider your previous and new (repeated) comments. We added explanatory information in the Legend of Table 3 (highlighted in green). .
It is really not possible to calculate "p-values" for the different parameters or in relation to age. For your information, please have a look at Table 2 in the attached pdf, of if that does not work, please have a look at : The Cow Milk Symptom Score (CoMiSSTM) in presumed healthy infants. Vandenplas Y, Salvatore S, Ribes-Koninckx C, Carvajal E, Szajewska H, Huysentruyt K. PLoS One. 2018 Jul 18;13(7):e0200603. doi: 10.1371/journal.pone.0200603.
Table 2 contains exactly the same information for different European countries as Table 3 in the paper on Egyptian infants. Since the paper you are reviewing contains only data from one country, we cannot do statistics for different countries.
The data presented in Table 2 in the Plos One paper and Table 3 in the paper you are reviewing are in fact "numerically presented" percentile curves. However, in order to make percentile curves over 2000 "inputs" are needed. (We have now data from ~ 2500 infants world-wide, and are making these percentile curves). So, this information is similar to growth-charts in children. It make statistically no-sense to show that a P3 weight-growth chart differs from a P25.
Moreover, statistical information is given in the manuscript just above Table 3:
The median (Q1; Q3) CoMiSS was 5 (5;6) (Fig. 1). The 95th percentile was 7. None of the infants had a CoMiSS of ≥10. There was no significant difference in CoMiSS according to gender (p=0.626) (Suppl. Fig 1), exclusively breastfeeding (p=0.198) (Suppl. Fig. 2), breastfeeding (total: mixed + exclusively) (p=0.430), formula feeding (p=0.886), or when complementary feeding was started (p=0.202).
A significant difference was seen in CoMiSS in the first age category analysis (p=0.02) (Fig. 2), but not in the second age category analysis (p=0.840) (Fig. 3). A Median CoMiSS of 5 was seen in all age categories (Table 3). A higher 25 percentile of 5 was seen in the 4-6 months, 8-10 months, and 10-12 months old group compared to a 25 percentile of 4 in the other age categories.
Please have also a look at the very recent papers in the European Journal of Pediatrics
- Cow's Milk-related Symptom Score (CoMiSS) values in presumed healthy European infants aged 6-12 months: a cross-sectional study. Jankiewicz M, Ahmed F, Bajerova K, Carvajal Roca ME, Dupont C, Huysentruyt K, Kuitunen M, Meyer R, Pancheva R, Koninckx CR, Salvatore S, Shamir R, Staiano A, Vandenplas Y, Szajewska H. Eur J Pediatr. 2024 Feb;183(2):707-713. doi: 10.1007/s00431-023-05334-0. PMID: 37973640; PMCID: PMC10912251.Table 2 represents similar data; no p values.
- The evolution of Cow's Milk-related Symptom Score (CoMiSS™) in presumed healthy infants. Bajerova K, Hrabcova K, Vandenplas Y.Eur J Pediatr. 2024 Jul 30. doi: 10.1007/s00431-024-05693-2. ) In Table 5, CoMiSS-Percentiles of different manuscript are listed, but no statistical analysis is done because it would be inappropriate to do so.
We do hope that this explanation clarifies the meaning and interpretation of Table 3.

Round 3
Reviewer 2 Report
Comments and Suggestions for Authors
Dear Authors,
As for Table 3, it was my mistake. What I meant was to put all p values which you discuss in 3.2 subchapter in additional table. As for the weaknesses of your work, admit, in the perfect world, having unlimited resources, wouldn’t you confirm your results and calculate correlation having CMA results? If so, add this thought to your work.
Author Response
Dear reviewer,
We thank you for you positive comments and suggestions.
What I meant was to put all p values which you discuss in 3.2 subchapter in additional table.
We did add a "Supplementary Table 3" with all the p-values mentioned in the subchapter 3.2 (highlighted in blue) .
As for the weaknesses of your work, admit, in the perfect world, having unlimited resources, wouldn’t you confirm your results and calculate correlation having CMA results? If so, add this thought to your work.
We did add a sentence at the end of the discussion - limitation section stating " It will be important to determine and compare CoMiSS values in infants with diagnosed CMA according to their ethnic origin, age and other variables." (also highlighted in blue)